# Extensive Orchards in the Agricultural Landscape: Effective Protection against Fraying Damage Caused by Roe Deer

**Petr Marada [1,2], Jan Cukor [1,3,*], Rostislav Linda [3], Zdeněk Vacek [3], Stanislav Vacek [3] and František Havránek [1]**

1   Forestry and Game Management Research Institute, v.v.i., Strnady 136, 252 02 Jíloviště, Czech Republic
2   Faculty of AgriSciences, Mendel University Brno, Zemědělská 1, 613 00 Brno, Czech Republic
3   Faculty of Forestry and Wood Sciences, Czech University of Life Sciences Prague, Kamýcká 129, 165 00 Prague 6 Suchdol, Czech Republic
*   Correspondence: cukor@fld.czu.cz

**Abstract:** The objective of this research was to determine the efficiency of different types of protective barriers and how they protect against fraying damage in extensive fruit tree orchards. Orchards in open agricultural land are the target of fraying damage caused by roe deer (*Capreolus capreolus* L.). We assessed the effectiveness of four protective barriers: a rabbit-proof fence, a standard plastic tube commonly used in forestry, and an innovative plastic tube—variants with and without an additional rendering fat application. The study was situated in three extensive orchards in the southeastern part of Moravia in the Czech Republic. We analyzed the ratio of damaged trees, stem circumference damage, the length and height of damage on tree stems, the time periods with the most observed damage, and finally, the economic efficiency of each studied barrier. Most of the damage was observed in April and July. The most effective protective barrier was the innovative tube with rendering fat application (up to 100%) followed closely by the innovative tube without rendering fat application (95%). The standard plastic tube had an effectiveness of 49%, while the rabbit-proof fence was the least effective at 25%. In terms of the mean damage-lengths on tree stems, we found no significant differences between the rabbit-proof fence and the standard plastic tubes (21–22 cm). The usage of the innovative plastic tube without rendering fat reduced the average damage-length by half (10 cm) as compared to standard types (rabbit-proof fence, standard tube) of protection. The damage-heights on tree stems showed no significant differences among all variants (53–58 cm from the ground). Our analysis of economic parameters showed that rabbit-proof fencing had the worst cost efficiency, while the innovative tubes without rendering fat, had the best cost efficiency. We recommend starting the installation of protective barriers on trees in March, since we recorded relatively high activity of male roe deer in the following months.

**Keywords:** fruit trees; extensive orchards; *Capreolus capreolus*; individual protection; pest management; Central Europe

## 1. Introduction

In the last centuries, agricultural landscapes have provided suitable microhabitats for many plant and animal species due to their diversity, the mosaic of landscape patches, or crop diversity [1,2]. However, the intensification of agriculture along with the declining number of cultivated species in the landscape have led to dramatic biodiversity loss and an overall homogenization of the landscape in the last few decades [1,3]. Worldwide soil degradation is another negative effect of agricultural intensification [4]. Sustainable management of agricultural soils and sustainable production are crucial

to reverse the trend of soil degradation which could lead to the desertification of actively managed agricultural land [5,6]. The vegetative coverage found in orchards also protects the soil against erosion therein supporting long-term sustainability of the agricultural landscape [6]. One option to increase landscape diversity is to plant solitary trees or establish extensive orchards, which has been found to affect biodiversity (mainly of weed species) [7–9]. Increased plant biodiversity is usually accompanied by an increased abundance of small game species [10–12]. Agricultural management practices also directly affect bird populations [13] which are dependent on plant diversity and insect species.

When establishing extensive orchards or planting solitary trees in an open agricultural landscape, damage to young trees has been encountered by ungulates. Ungulates have been able to adapt to agricultural intensification and their population has been rapidly increasing throughout Europe [14–16]. Damage on solitary trees is caused mainly by roe deer (*Capreolus capreolus*). The abundance of roe deer has been on the rise in the Czech Republic as well (according to numbers of hunted individuals or counted numbers of game) [17].

Therefore, planting extensive orchards needs to be protected, especially from the territorial behavior of male roe deer (damaging of tree stems by fraying), which could cause significant economic losses [18]. In open agricultural landscapes, orchards sustain damage more frequently by roe deer due to the lack of forest coverage, or game refuges with trees and shrubs, which are typically used by male deer for fraying and marking of the territory [19]. Suitable protection of tree stems could significantly (or even totally) eliminate this damage and is crucial for the successful establishment of orchards. Although fencing an entire orchard could be effective [20], it is not desirable, as it reduces the permeability of the landscape for small game, it is quite expensive [21,22], and is not viable for many owners [22].

Chemical or mechanical protective barriers (e.g., repellents) could be alternative methods to fencing. Protection using chemicals has often been observed to have low effectiveness in comparison to other methods. Natural preparations, which are usually composed of active ingredients like denatonium benzoate, capsaicin, putrescent whole egg solids, and other extracts [20,23,24], could reduce stem damage for several weeks, but generally lose their effectivity after a few months [23,25]. Common mechanically-based protective barriers have been stem covers that are designed to prevent bark stripping from the trunk. Although such covers are already widely used by foresters, these barriers can be destroyed by fraying which makes tree stems prone to further damage.

Complete, effective protection of fruit trees in orchards (except for fencing an entire orchard area) seems to be challenging. Therefore, we designed a user-friendly (it allows the removal of annual shoots growing from base and the stem of young fruit trees), plastic protection tube, which protects against fraying and bark stripping, and does not hurt wild game.

The aim of this study is to (1) determine the best application time for stem protective barriers based on year-long damage distribution observations in locations with high population density of roe deer, (2) to compare the effectiveness of innovative stem protection tubes with standard tubes typically used in forestry, and (3) to evaluate the economic efficiency of the selected protection types.

## 2. Materials and Methods

### 2.1. Study Area

The study was conducted in three permanent research plots (PRPs) in the southeastern part of Moravia (Czech Republic) within the cadastral territory of Šardice, as shown in Figure 1.

According to hunting and game management, all three permanent research plots belong to the hunting district of Šardice, as shown in Figure 1, with an acreage of approximately 1833 ha. The composition is as follows: agricultural land = 1666 ha (90.9%), forests = 83 ha (4.5%), aquatic areas = 8 ha (0.4%), and other areas (roadways, recreational areas, urban greenery, field roads, etc.) = 76 ha (4.1%). The only ungulate game species present in this district are roe deer (*Capreolus capreolus*) and approximately 52 individuals have been hunted per year (average from 2013–2017).

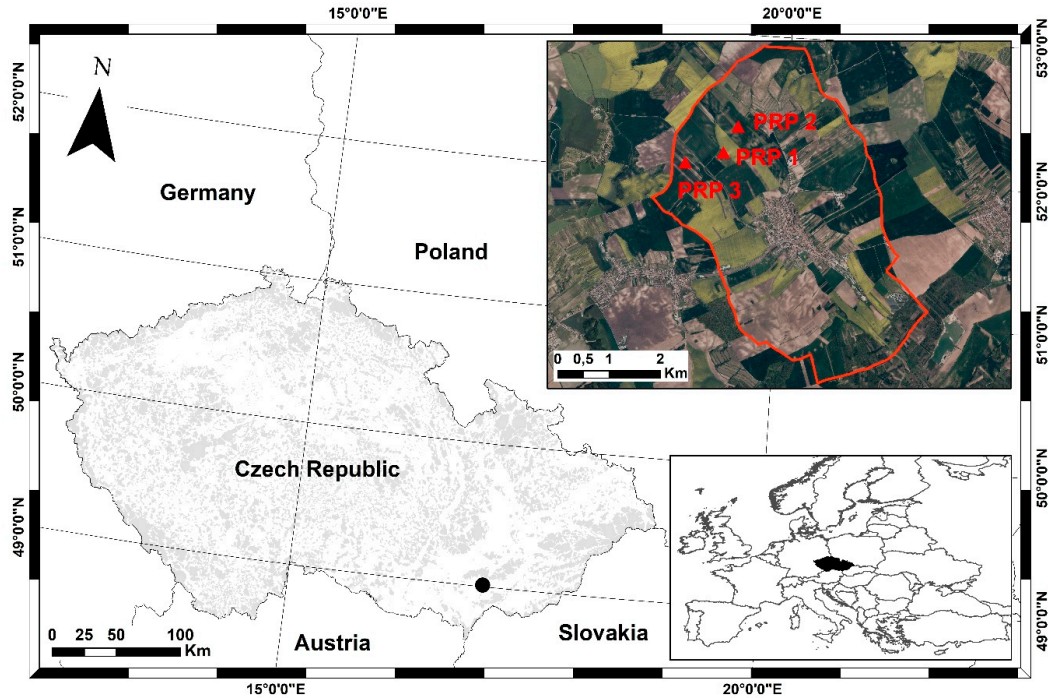

**Figure 1.** Localization of permanent research plots (PRPs) (symbol ▲) within the hunting district of Šardice (symbol ●) in the southeastern part of Moravia (Czech Republic). Grey areas in the map depict forested areas in the Czech Republic. [Underlying data sources: © Czech State Administration of Land Surveying and Cadastre, ©ArcČR, ARCDATA PRAHA, ZÚ, ČSÚ, 2016].

## *2.2. Orchard Description*

Extensive orchards were established on all three PRPs in 2015. Fruit trees were planted in 8 × 6 m rectangular areas, so that the average number of trees per hectare was around 50. The number of trees planted in each PRP are presented in Table 1. Altogether, there were 400 fruit trees planted within the three PRPs with a mean tree diameter of 49.6 mm ± 10.95 SD.

**Table 1.** Overview of basic characteristics of permanent research plots.

| PRP | GPS Coordinates | Acreage (m$^2$) | Altitude (m) | Number of Trees | Type of Protection | | | |
|-----|-----------------|-----------------|--------------|-----------------|--------------------|---|---|---|
| | | | | | Rabbit-proof fence | Standard plastic tube | Innovative plastic tube | Innovative tube with rendering fat |
| 1 | 48°58′38.7″N 17°0′31.1″E | 44,447 | 211 | 250 | 75 | 75 | 50 | 50 |
| 2 | 48°59′0.8″N 17°0′34.2″E | 19,843 | 242 | 100 | 25 | - | 25 | 50 |
| 3 | 48°58′39.8″N 16°59′55.1″E | 10,124 | 219 | 50 | - | 50 | - | - |

For each species, 64 individuals were planted. This included European pear (*Pyrus communis* L.), apple (*Malus domestica* Borkh.), apricot (*Prunus armenica* L.), sour cherry (*Prunus cerasus* L.), except for plum (*Prunus domestica* L.) and sweet cherry (*Prunus avium* L.), where 72 individuals were planted.

## 3. Methods of Protection

Two standard and commonly used methods were selected to protect fruit trees against fraying damage caused by male roe deer: rabbit-proof fencing and plastic tubing. Rabbit-proof fencing consists of light wiring with a small size of mesh—hexagonal, 13 × 13 mm, wire diameter of 1 mm. Better resistance to climatic conditions was ensured by adding a PVC surface finish to the rabbit-proof fence.

Standard plastic tubes are mostly used to protect broadleaved tree species against damage from game (mainly against browsing) in forestry and were selected. The plastic tubes were 120 cm high with a squared floor plan and an internal diagonal of 8.5 × 8.5 cm. Tubes were made of polypropylene, with the stated lifespan of such material between 3 and 4 years.

Based on previous negative feedback from landowners with insufficient orchard tree protection, an innovative type of tube protection was designed and evaluated with respect to efficiency. The innovative plastic tube was specifically designed for orchard tree protection in open agricultural lands. The height of the innovative plastic tube was also 120 cm, following the standard tube height commonly used in forestry. The shape was conical, with a bottom diameter of 12 cm, and a top diameter of 10 cm (for more detailed technical information of the innovative plastic tube, see Figure 2). The chosen material was elastic and not prone to degradation. We predict the lifetime expectancy to be longer than 10 years, after which time the stem of orchard trees should no longer be so attractive to male roe deer. Another difference from the standard plastic tube was the ability to open the innovative tube and remove the shoots from the stem. A second variant was created with rendering fat capability, which was applied to the bottom-most section of the cover, approximately 40–70 cm from the ground. The trees selected to be covered in protective barriers were selected randomly on each plot.

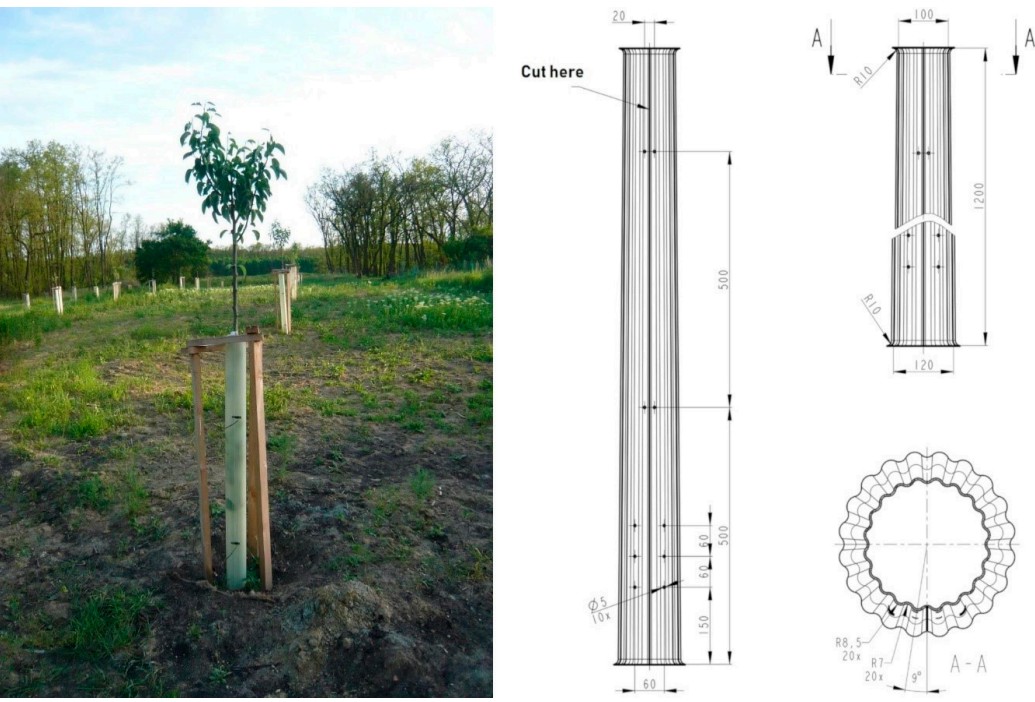

**Figure 2.** The innovative plastic tube in application (**left**) and its technical parameters (**right**). Rendering fat, when used, was applied on the bottom part of the cover.

### 3.1. Evaluation of the Protection Efficiency

For the evaluation of protection efficiency, control measurements of tree damage were conducted after three-year periods. The width and height of tree bark damage caused by fraying behaviors of male roe deer was measured while other damage was not observed. The width and length of damage was measured at the site with the heaviest observable damage on the stem using a standard forestry tape measurer in millimeters. The height from the center of the damaged area on each stem was also measured. Stem circumference was measured at the height of 50 cm (site on the stem where fraying damage was observed in most cases). Every PRP was monitored each month within the three-year period to evaluate the extent of damage present. Based on measurements found, the season with the highest risk of fraying damage was identified, which has important implications for the effective installation of orchard tree protective barriers.

The time necessary for the installation of each type of barrier was measured while the orchard was established in order to assess the financial cost of tree protection services. The total cost was calculated as such: material cost [EUR] + application time [h] × hourly wage [EUR]. Average hourly wage of auxiliary workers in forestry (5.03 €) was determined according to the Average Earnings Information System in the Czech Republic (ISPV; www.ispv.cz). The conversion of the currency (from CZK to EUR) was based on the Czech National Bank exchange rate (as of 20 March 2019).

### 3.2. Data Analysis

The differences exhibited in damaged tree ratios, stem damage circumference ratios, mean heights of stem damage centers, and mean damage lengths on the stems were evaluated between the selected types of protective barriers (rabbit fence, standard plastic tube, innovative plastic tube without rendering fat, innovative plastic tube with rendering fat). With respect to stem damage circumference ratios, separate analyses for each and every damaged specimen are provided. To evaluate the differences in damaged tree ratios between selected levels we used the Pearson chi-squared test, as well as the Agresti et al. [26] method for multiple comparisons. The differences in the remaining three parameters were evaluated by the Kruskal–Wallis test, followed by relevant multiple comparisons. All statistical procedures were conducted in R software [27]. The significance level was set to $\alpha = 0.05$. The radar chart displaying multivariate data (time of installation, cost of protective barrier, expiration of protective barrier, parameters of damage on tree samples) was used to evaluate the total efficiency of individual types of protection. We used a simple bar plot to present an illustration of damage that occurred as well as distribution through the years.

## 4. Results

The differences in the ratio of damaged trees between selected types of protection was tested via the Pearson chi-squared test (chi-squared = 124.9, df = 3, $p < 0.001$). Subsequent multiple comparisons showed significant differences between the original methods (rabbit-proof fence and plastic tubes) and the innovative plastic tube methods (innovative tube with or without rendering fat), as well as between the rabbit-proof fence and the standard plastic tubes, as shown in Figure 3. Damaged tree ratios were substantially lower when the innovative plastic tubes were used and, notably, when the innovative plastic tubes with rendering fat were used, no damaged trees were observed.

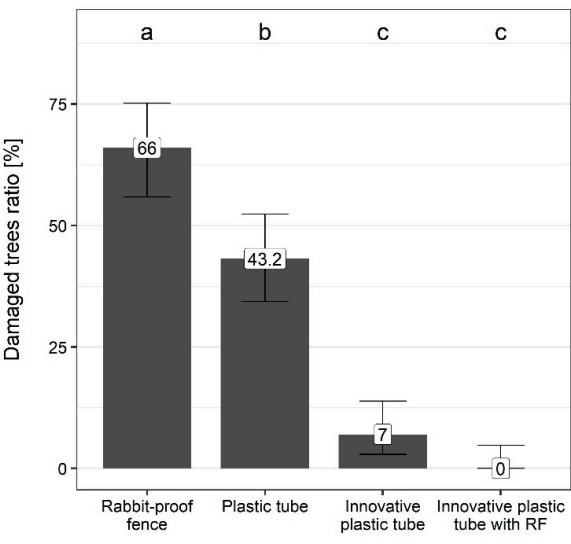

**Figure 3.** Overall ratios of damaged trees for selected protection types. Error bars depict 95% confidence intervals. The letters above bars represent statistically significant differences between variants (significantly different variants have different letters above respective bars).

Damaged stem circumference ratios had very similar results. The Kruskal–Wallis test showed significant differences between selected types of protection when all studied trees were involved in the analysis (chi-squared = 124.2, df = 3, $p < 0.001$). Marginal insignificance was found only when damaged trees were analyzed (chi-squared = 5.16, df = 2, $p = 0.076$) as shown in Figure 4. The innovative plastic tubes significantly restricted the extent of the damage. When the tree was damaged, on average, only 19.8% of stem circumferences were affected by fraying. When using plastic tubes, the values were 64% higher, and rabbit-proof fencing exhibited double that value, as shown in Figure 4A. When the circumference damage was compared to all studied trees, only 1.4% of the total hypothetical tree stem circumference was damaged when using innovative plastic tubes. Standard plastic tubes showed an increase of this value tenfold and rabbit-proof fencing almost twentyfold, as shown in Figure 4B. Stem damage led to tree mortality in nine cases, all of which were protected by the rabbit-proof fence.

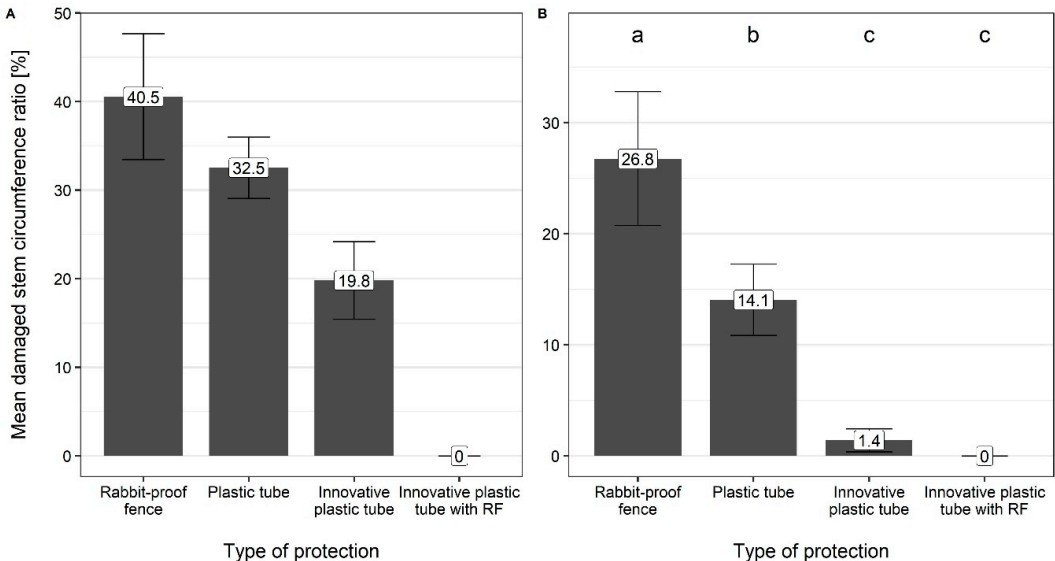

**Figure 4.** The comparison of mean damaged stem circumference ratio for damaged trees (plot **A**) and all studied trees (plot **B**). The error bars represent a 95% confidence interval. The letters above bars represent statistically significant differences between variants (significantly different variants have a different letter above their respective bar).

The comparison of mean damage length also showed a similar pattern as in the previous analyses. The Kruskal–Wallis test pointed out statistically significant differences between mean damage lengths for selected variants (chi-squared = 7.71, df = 2, $p = 0.02$), as shown in Figure 5. No significant differences were found between the rabbit-proof fence and the standard plastic tube. On the other hand, innovative plastic tubes showed an approximate average of only half of the length of fraying damage compared to the standard types of protection.

When evaluating damage height (on the stem), no significant differences were found. The mean height with respect to the center of stem damage was observed at 54.2 cm from the ground on trees protected by the rabbit-proof fence, at 52.8 cm from the ground on trees protected by plastic tubes, and at 58.0 cm from the ground on trees protected by innovative plastic tubes. No fraying damage was found for innovative plastic tubes with rendering fat application.

The distribution and relative occurrence of damage observed throughout the study is depicted in Figure 6. For months that are not presented in the plot, no damage was recorded. The least amount of damage was observed in March or August of each recorded year (ca. up to 8% in both months in total for each year of the study). Generally, most of the damage was recorded in April or July—over 30% of total damage occurred in July of each year. Over 40% of damage recorded in 2013 happened in April, with lower values, ca. 23% in 2014 and 2015, recorded in the following years.

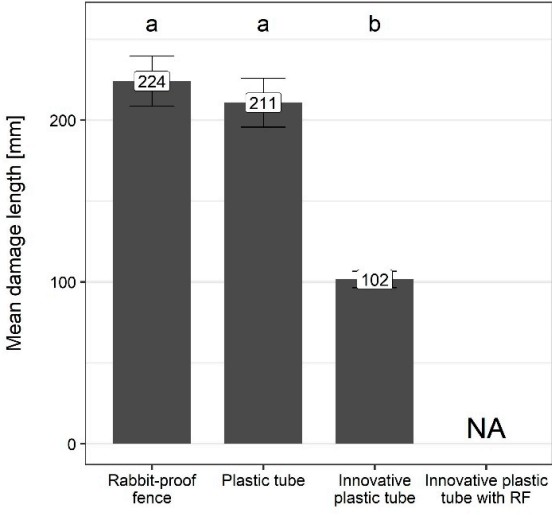

**Figure 5.** The comparison of mean stem damage length for damaged trees. Error bars represent mean ± SE. The letters above the bars depict statistically significant differences between variants (significantly different variants have a different letter above their respective bars).

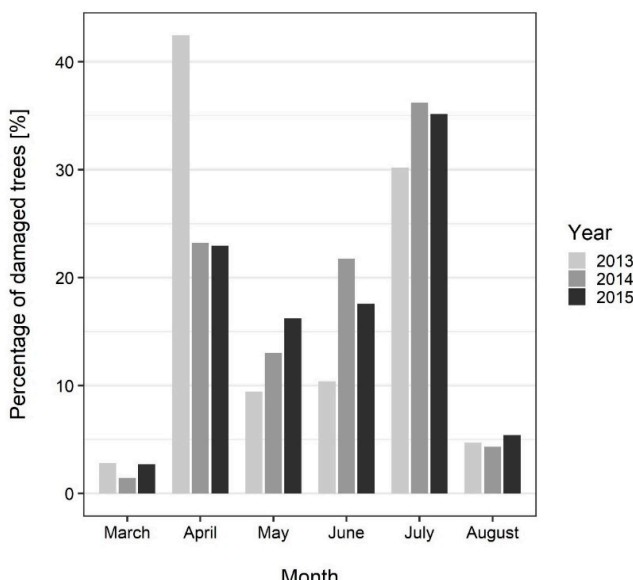

**Figure 6.** Relative distribution of damage occurrence throughout each year of the study.

The total efficiency, including the cost of each protective barrier, time taken for installation, expiration date of protective barrier, and parameters of damage found on tree stems was observed. According to our observations, the most effective was the innovative plastic tubes with rendering fat application, followed by innovative plastic tubes without rendering fat, while the rabbit fencing and standard plastic tubes were the least efficient, as shown in Figure 7. The installation times were recorded as follows: rabbit fence = 19 min, innovative plastic tube with rendering fat = 7 min, innovative plastic tube = 5 min, and standard plastic tube = 5 min. It is noteworthy to state that 96 pcs of plastic tubing or innovative plastic tubing can be installed in one day shift compared to 25 pcs of rabbit fencing. The most expensive type of protection was the rabbit-proof fence at 4.51 € (2.95 € material), while the least costly protection was the standard plastic tube—1.66 € (1.24 € material). The innovative plastic tube with rendering fat cost an average of 2.34 € (1.75 € material) and the innovative plastic tube without rendering fat cost 1.97 € (1.56 € material). Both types of plastic tubing had the longest expiration dates, while the rabbit fence had the shortest expiration date.

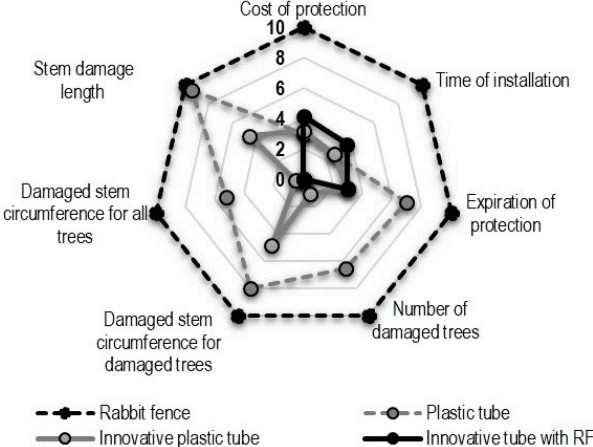

**Figure 7.** Efficiency assessment of individual types of protection; a relative value of 10 in the radar chart indicates the lowest efficiency = maximum value in each parameters of protection types; lower surface = greater efficiency.

## 5. Discussion

In recent decades, the conventional agricultural landscape has been characterized by intensification, homogenization of the farming landscape according to time and space, and a higher usage level of chemicals like pesticides or fertilizers [1,8,9,28]. Many environmental risks have been related to the use of conventional insecticides e.g., their aerial dissemination and the contamination of soil or water negatively affects many wildlife communities [9]. Such changes in the agricultural landscape have been one of the main causes for the declination in diversity and abundance of wildlife species [28–30]. One option to support heterogeneity, biodiversity, and long-term sustainability of an open agricultural land is to establish extensive orchards. Unfortunately, in Europe, these areas are often completely fenced to prevent damage caused by wildlife [31]. However, completely fencing an entire orchard area has been shown to be useless if extensive orchards should fulfill their environmental function to assist permeability for some animal species. Extensive orchards could prevent biodiversity loss and also serve as a biodiversity hotspot in the landscape. It has been shown (in peach orchards, for example) that extensive orchards could provide suitable habitat for up to 70% more biodiversity in comparison to conventional agricultural systems [32]. The evaluation of potential risks to these habitats caused by anthropogenic activity needs to be continuous in order to assess the sustainability of used agricultural practices [33].

A lot of attention has been focused on protecting not only fruit trees in orchards and alleys as well as other tree specimens in open agricultural land but also in forestry [34–37]. Orchards sustain the most damage from roe deer, whose preferred habitat includes forest edges and woodland steppes, but have also successfully adapted to live in open agricultural lands [38,39]. The latter behavior has been observed since the beginning of the 20th century and, now, it is possible to divide the roe deer population into forest and field populations based on habitat preferences [38]. In open agricultural lands (where shrubs are absent), the presence of branches or other forest vegetation in the area have strongly affected damage intensity on horticulture. For example, data from the United Kingdom shows that 84% of damage sustained in horticulture and orchards has been caused by roe deer [40].

Our results show significantly higher efficiency of both variants of innovative plastic tubes (with or without rendering fat) compared to commonly used methods especially in comparison to rabbit fencing. The standard plastic tube has been most commonly used in forestry to minimize browsing damage. In terms of fraying damage caused by roe deer, the standard plastic tube has been less effective from the point of lower material stability and early degradation.

Fraying damage has been a crucial factor in the implementation of extensive orchards and has been found to lead to fruit tree mortality (2.3% of trees involved in the experiment died because

of fraying damage). The occurrence of fraying damage has been the condition of Central Europe and has been associated with defending the territory of adult roe deer males from the early spring (August) until the end of the mating season in the summer [19,41,42]. This statement corresponds with our findings, where the initial damage peak was observed in August (territory defending) with the highest probability of fraying in the mating season (July). On the other hand, other types of damage (browsing, bark stripping) have often been affected by the feeding preferences of a deer and has been most problematic in the winter season as fodder sources are limited [20].

It is possible to explain the high level of efficiency (up to 100%) exhibited by the innovative tubes with rendering fat by territory marking behavior. Roe deer bucks usually mark trees, shrubs, and branches with a specific scent that is typical not only for older males, but also for younger males [19,43]. Remarking the scent made by other individuals is also a common practice [43]. However, when rendering fat had been applied directly on the plastic tube, the fruit tree was no longer attractive for scent marking. Other repellent applications used for the protection of trees in orchards showed different amounts of efficiencies [20,23,25,44]. Concrete data based on a survey was published by Lemieux et al. [24], where no or slight efficiency was declared in 49% of cases, moderate effectivity was found in 39% of cases, and high effectivity was reported only in 12% of cases. Generally, it is more efficient to combine repellents with the use of a mechanical protection.

We have shown a suitable solution to protect fruit trees in extensive orchards against fray damage caused by roe deer, which is the most widespread ungulate in open agricultural land of Central Europe [45]. The innovative plastic tube has shown a higher level of convenience from its cost point, installation time, and expiration date, all of which are more favorable when compared to standard plastic tubes. In order to assess the application of each type of protection, it is necessary to take into account the local experience with habitat conditions, the character of extensive orchards, and the game management within each particular hunting district [46]. To regulate this high level of efficiency, it is highly recommended to adhere to the technology of the application and maintain the protective barrier throughout the year [47]. To systematically reduce fraying damage, it is also necessary to improve the management of the large herbivore populations [39,40,46,48,49].

## 6. Conclusions

Extensive orchards in open agricultural lands have been characterized by low rates of agricultural intensification and biodiversity richness as well as higher ecological values. However, if orchards are to play their full extensive role, they should be effectively protected against damage caused by the ungulate game species. Based on the increasing population of roe deer in Central Europe, an innovative type of plastic tube was designed and used for individual protection of fruit trees.

The innovative plastic tube shows significantly greater efficiency (up to 100%) against fraying damage compared to other commonly used types of protection and provides an alternative to fencing an area. The innovative plastic tube has shown to be both suitable and applicable for establishing a territorial system of ecological landscape stability and planting solitary trees or an extensive orchard and/or other greeneries, especially where the complete fencing of an area would not be a suitable solution to protect trees. Appropriate protection of fruit trees or other solitary trees could ensure their successful growth and preclude their mortality.

**Author Contributions:** Conceptualization, P.M., J.C. and F.H.; methodology, J.C., R.L. and F.H.; software, R.L.; validation, P.M., S.V. and F.H.; formal analysis, J.C., R.L. and Z.V.; investigation, J.C.; resources, P.M.; data curation, J.C., R.L and Z.V.; writing—original draft preparation, P.M., J.C., Z.V. and S.V.; writing—review and editing, J.C., S.V. and Z.V.; visualization, R.L.; supervision, P.M.; project administration, P.M.; funding acquisition, J.C. and P.M.

**Funding:** This study was supported by the Czech National Agency for Agricultural Research, project number: QJ1530348 and also by the Czech University of Life Sciences in Prague, Faculty of Forestry and Wood Sciences, Internal Grant Agency, Project No. B 19/05.

**Acknowledgments:** We thank Czech National Agency for Agricultural Research and Czech University of Life Sciences in Prague, Faculty of Forestry and Wood Sciences for providing funding for this study. Our thanks belong also to anonymous reviewers and editors for improving our study during peer-review process.



**Conflicts of Interest:** The authors declare no conflict of interest.

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
