# Peer review of "Extensive Orchards in the Agricultural Landscape: Effective Protection against Fraying Damage Caused by Roe Deer"

_sustainability, doi:10.3390/su11133738_

Round 1

Reviewer 1 Report

Review for

 Extensive Orchards in Agricultural Landscape: Effective Protection Against Fraying Damage Caused by Roe deer

This paper has some great evidence of what type of protection is best used for extensive orchards in order to keep down damage from ungulates. However, it is in need of extensive editing to make the English grammar correct.  It also lacks an appropriate amount of information and methodology that relates to sustainability. This article would be much better appreciated in an agriculturally related journal. I therefore recommend rejecting this article for the journal Sustainability as it does not relate to the aims/scope of the journal.

Edits suggested are formatted as Lines: suggestion

23: remove the word “of”

27: remove the word “found”

28: you wrote “in average a half lower values”, correct grammar would be “on average values were reduced by half”

31: “efficiency was observed in case of” insert “the” so it says “observed in the case of”

32: “innovated tube without rendering fat. Although the efficiency of innovated tube with rendering fat turned out to be very high, it is recommended to begin”, should be changed for a better flow, I recommend: ““innovated tube without rendering fat, although the efficiency of innovated tube with rendering fat turned out to be very high. We recommend to begin”

36/37: You need to list at least 3 keywords

Grammar issue lines: 43, 44, 60-62, 64, 72-75, 86, 97, 98, 139, 142, 150, 175/176, 181, 187, 188, 190, 198, 202, 214/215, 218, 225, 227, 228-230, 244-248, 254, 255, 257, 264, 266, 270, 273, 282, 286, 288, 294/295,

Most grammar issues are with the use of the, on, of, in (prepositions), some are sentence structure.

53: I recommend only putting parenthesis around (UHUL 2017) and not half of the sentence.

72-75: this sentence is too long and needs to be broken up. Also there is no need to start a new paragraph in line 76 otherwise these lines are too few for consideration of a paragraph.

80: you could also add a sentence on what you hypothesize will be the best option and why.

83-93: I don’t think the specifics of the average weather is important to your story about deer (unless you correlate deer and weather later in the paper) therefore this paragraph can be shortened.

99: I like this figure, but if there is a way to make the aerial photo larger than the map of Czech Republic I could see the differences in landcover type better.

108: size is in mm and usually I see size in cm, the choice is yours.  Also, a tree is not a piece so I would not write “pieces of European pear” ect.

109: correct English would be “64 European pear (latin name), 64 apple (latin name), 64 apricot (latin name) trees were planted.” It would also be correct to write it as such: “For each species 64 individuals were planted, this included European pear (latin name), apple (latin name), apricot (latin), sour cherry (latin), except for plum and sweet cherry where 72 individuals were planted.”

125-133: was instead of is

133: how much of the bottom of the tube was covered? Please state how many centimeters from the bottom it was applied to.

144: remove “also three years after plantation”, this was already stated

145: I don’t understand what you mean by controlled twelve times.

153: I don’t know what ISPV is but double check you cited correctly.

154: exchange rates vary so please in parentheses put the timeframe you used for the exchange rate.

162: the Agresti et al 2008 is for citation so put [25] next to it.

168: year should be years

184: “For results of multiple comparisons, see Fig. 4.” This sentence can be removed and you can write (Fig. 4) at the end of the sentence is line 183

186-187: cite (Fig. 4a) for these sentences

188-192: cite (Fig. 4b) for these sentences

194 & 205: This does not have to be bold letters.

210: I do not think you need a graph for these results.  I would just explain them in another sentence giving their means per treatment.

279/280: This sentence is confusing, what particular application are you referring to?

300-303: sentence is too long.

Author Response

Dear Reviewer 1,

     thank you for your beneficial comments. All your suggestions were answered and the whole manuscript was edited by native English speaker.

On the behalf of authors collective

                          Jan Cukor

Reviewer 2 Report

This is a simple but very clear study, with relevant implications for the management of damages on fruit trees provoked by ungulates in agricultural landscapes. The ecological and management framework, and antecedents, of the study problem, are very well stated. The relevance of the objective is well-founded; so that forest agroecosystems can generate their multiple benefits (production and habitat for biodiversity), it is necessary to develop effective methods to protect young trees from ungulates, without enclosing the farms. The experimental design to evaluate the performance of 4 methods is correct, but more details are needed to better understand the methodology. The statistical is simple but adequate. The results are clearly presented, analyzed and interpreted. The main conclusions are supported by the study results. I have detected some language details despite I am neither an English-speaker.

 Recommendations

1)      Methodology: Explain how the different experimental treatments were assigned to individual trees. Did you use a completely random design? Did you include as an experimental factor the tree species?

2)      Make check the English by an English-speaker

 Some minor revisions

1)      Include keyword

2)      Language revision, for example:

Lines 91-92: “The bedrock in the locality consist mostly of marl”. Change to “consists”.

Lines 130-131: “Another difference against standard plastic tube is possibility to…” Change to “is the possibility”

Lines 149-141: “The width and the length of damage was measured”. Change to “….were measured”.

Lines 188-189: The sentence is not clear.

Lines 243-247: The sentence is not clear

“Innovated tube”. Change to “Innovative tube”

Author Response

Dear Reviewer 2,

     thank you for your beneficial comments. All your suggestions were answered and the whole manuscript was edited by native English speaker.

On the behalf of authors collective

                          Jan Cukor

Reviewer 3 Report

Described solution of innovated plastic tube in the manuscript showed significantly greater efficiency (up to 100 %) against fraying damage compared to commonly used types of individual protection. This Innovated plastic tube enables its applicable especially for  planting of individual trees or establishment of orchards or other plots, where complete fencing of area is not suitable solution of mechanical protection of trees. The problem of tree protection against animals it crucial in many case, especially in agriculture and forestry.
In the manuscript was prepared properly and adequate research methods are applied.
The research aim clearly defined and answered.
The key words were not prepared - it is suggestion to add the keyword to made the paper more recognise for adequate search engine scientific data base.

Author Response

Dear Reviewer 3,

     thank you for your comments. The information about height of rendering fat application was added into the text. There were also added the keywords into the Sustainability submission system and the whole manuscript was edited by native english speaker.

On the behalf of authors collective

                          Jan Cukor